# Subsidiarity Principle—Its Realization in Self-Government in Lithuania and Poland

**Robert Gawłowski** [1,*] **, Saulius Nefas** [2] **and Krzysztof Makowski** [1]

1   Finance and Management Department in Bydgoszcz, WSB University in Torun, 87-100 Torun, Poland; krzysztof.makowski@wsb.torun.pl

2   Public Management Innovation Laboratory, Romeris University, Vilnius 08-300, Lithuania; saunef@mruni.eu

*   Correspondence: robert.gawlowski@wsb.bydgoszcz.pl

**Abstract:** Purpose: This paper analyzes the development of the concept of subsidiarity, its relationship with management theories and the implementation of the principle of subsidiarity in the context of local self-government in Poland and Lithuania by conducting comparative research. Design/ methodology/ approach: The research was conducted based on methods of desk research of scientific literature, analysis of documents and their content from the Lithuanian and Polish parliaments' commissions and using comparative approach methods. Findings: The research reveals the essential aspects of the concept of subsidiarity and the extent to which this principle is relevant in the work of the Committee on State Administration and Local Authorities of the Seimas of the Republic of Lithuania and Self-Government and Regional Policy Commission of the Sejm of the Republic of Poland in shaping the policy of self-government. The study demonstrates that subsidiarity principles are used in practice in a very incoherent way regarding public service regulations. Practical implications: The analysis is relevant, as an implementation of public policy in Lithuania and Poland increasingly calls for the integration of values into politics, which cannot be done without relying on certain principles. Research limitations: The generalizability of the results is limited by the number of parliamentary terms that have been taken into consideration. Originality/ value: The originality of this paper lies within the context in which this study took place—an international comparison of Lithuanian and Polish concepts that is rarely taken into consideration in the scientific literature.

**Keywords:** subsidiarity; self-government; public administration

## 1. Introduction

Implementation of the principle of subsidiarity is a prerequisite of modern local self-governance because it complements, extends and discloses the principle of separation of powers from the managerial point of view.

Application of the principle of subsidiarity is extremely relevant in areas where the general competence of the entities of public governance comes into the foreground. We believe that this is directly connected to making decisions on local self-government or local community. We aim to establish whether implementation of the principle of subsidiarity in modern systems of self-government in Lithuania and Poland is substantive or declarative. The problematic questions to be addressed are the following: how do we perceive the principle of subsidiarity and do we use it in decision-making (while adopting legislation related to the local level)?

Thus, on the one hand, the advantage of subsidiarity is that it prevents central government entities from becoming too powerful, while on the other hand it helps to prevent confusion in decision-making and duplication of the same functions at different levels of government, as all levels of public governance are integral parts of public governance systems. To ensure the smooth functioning of the whole system,

it is important to promote coherence between actions of all subsystems. Therefore, bodies involved in public governance must adhere to the principle of coordination of interests while exercising their competence, which is indispensable to the principle of subsidiarity.

The aim of this article is to reveal the extent of the application of the principle of subsidiarity in the work of committees on local authorities in Lithuania and Poland when deliberating draft legislation directly related to local self-government and local community, by means of empirical research based on the analysis of the genesis of the concept of subsidiarity and its relevance in policy implementation of local self-government and local community.

This research relies on the methods of scientific literature analysis, document analysis and content analysis.

## 2. The Concept and Development of the Principle of Subsidiarity

The research starts with the analysis of the development of the concept of subsidiarity, as there are at least four reasons for that. First, we need to get a closer insight into the meaning of the concept rather than refer to it as some abstract form. In an abstract form we do not find or observe anything new, because apparently there is nothing new in it—according to Hegel (1997, p. 50), in modern times an individual finds a ready-made abstract form, and, as Jokubaitis (2016, p. 27) extends Hegel's thought, someone has performed the act of thinking for us and provided us with ready-made concepts and answers.

Secondly, after agreeing on the concept of subsidiarity to be used, we have to analyze its application in practice, and here we need individuals to understand it rather than constantly repeat it to look fashionable and modern without actually understanding what the concept of subsidiarity means. The analysis of the eight principles entrenched in laws of the public administration of Lithuania and Poland, among which the seventh one is subsidiarity, shows that only a few individuals can explain what subsidiarity mean The majority who try to explain it say that it is something related to financial subsidies. Thus, to understand not only these, but also other concepts, we need to know and perceive their nature, because translation of the word 'concept' has a number of meanings such as 'an abstract idea', 'principle' or 'notion'. Denominations of concepts are created by people, thus they are essentially arbitrary (De Castera 1994). However, knowing the denomination does not mean understanding the concept. Understanding the concept involves conceptual knowledge.

The third reason is that although we are not going to argue which formulation of the concept of subsidiarity is the best (that is the question often addressed by practitioners), by showing their abundance and different interpretations during the evolution of the concept, we want to highlight that it has been relevant at all times.

Finally, the fourth reason is related to management, as the application of this concept is crucial to the implementation of public policy and administration (from the local level to the highest level of management); thus we will try to contemplate which concepts of subsidiarity or their elements can be most comprehensible and applicable (Bieliauskaitė 2019).

## 3. Evolution of the Concept and Its Application

Analysis of the evolution of the concept should be started from antiquity, when, according to Aristotle (Vaišvila 2004, p. 88), subsidiarity meant the idea of prosperity for all with an emphasis on the organic relationship between the whole and differentiated parts within the state organization. Although this conception is rather abstract, in the sense of the subject, the relationship here is limited to the state.

The concept of subsidiarity in post-modern times is found in the documents of the European Union (EU; first in Maastricht Treaty of 1992), though it was used much earlier in informal discussions keeping with the traditions of Catholic social thought (Dromantienė and Česnuitytė 2011, pp. 15–16). However, as the authors say, the EU social policy, unlike countries of the Catholic tradition, distinguished the principle of subsidiarity from the solidarity principle which refers to assistance to a person or social

groups and puts more emphasis on the fact that external intervention is only possible when a Member State is unable to achieve certain goals and intervention actions are subsidiary only, while decisions must be taken at the lowest level possible, as close as possible to the citizens, in order to maintain their autonomy, that is, the goal of the collective action is an individual; however, no collective effort should do something that the person is capable of doing. Subsequently, the Treaty of Amsterdam 1997 (Popławska 2000, pp. 16–18). presents the principle of subsidiarity in such a way that action carried out by the State, in comparison with the action carried out by local authorities, would be much more beneficial because of its scale and effectiveness. Therefore, without any exaggeration we can say that the EU Treaties recognize the legal-political principle of subsidiarity and aim to organize the growing number of competences that are shared between the EU, the Member States and substate levels of government and it is a foundation block to multilevel governance on the European level (Pazos-Vidal 2019, p. 2). Despite the fact that the subsidiarity is one of the most important political principles in the EU, there is still vital discussion at the European level on how to bridge the gap between the ideal and the reality (Lopatka 2019).

Kondratien (2011) analysed the practical application of the principle of subsidiarity in a public organization, its impact on the levels of government in the state, attainment of the balance of power based on subsidiarity, and efficient distribution and implementation of functions in the discourse of the Lithuanian science; however, the object of her research within the state governance hierarchy was local self-governance, which, according to the law[1], is defined as self-regulation and self-activity of the community of permanent residents of an administrative unit, which has the right of self-governence guaranteed by the Constitution; she did not focus on a local community, which according to the same law refers to residents living in a neighbourhood related by the common needs and interests of living in that neighbourhood, who act through various forms of direct participation in order to meet these needs and interests. Although the terms "local" and "local community" are used (Kondratien 2011, p. 7), they are used as synonyms and not in the sense of functional or management levels. Besides, the priority of the author is given to the methodological possibilities of legal regulation of subsidiarity or analysis of subsidiarity as an interpretation of the law of social order (Kondratien 2011, p. 8).

In terms of the Polish context, the subsidiarity principle can be understood in relation to the public administration organization. Therefore, it is mentioned simultaneously with principles such as efficiency, unitary state and hardening local communities. It is always part of local and central government relations (Izdebski 2014, p. 39). Moreover, the subsidiarity principle is very often interpreted on an axiological basis to the political system. Therefore, it is not only part of public administration but has also been a main driver of political transformation since 1989 (Zaradny 2018, p. 11). However, it does not mean that public services should always be delivered by possibly the lowest level of public administration. The subsidiarity principle should be used by government as a one of the most important drivers in terms of public service division. (Krasnowolski 2012) Taking this fact into consideration, it is possible to centralize and decentralize public service organization during the time (Sarnecki 2005, p. 57). Moreover, there is no doubt that the subsidiarity principle has the same legal position as other constitutional ones such as division of powers or citizens' rights (Polskiej 1997).

Therefore, we can claim that the implementation of the principle of subsidiarity can take place in decision-making at several levels, that is, in interstate relations, within a state between different hierarchies of governance, and in relations between people, that is, the community and an individual (Regulski 2005). In this research, we aim to reveal whether this system functions, that is, whether there is a managerial mechanism to support it. Therefore, it is logical that we need to find out how representatives of management science highlight the functioning of subsidiarity in their theories.

---

[1] Lietuvos Respublikos vietos savivaldos įstatymas. Lietuvos Respublikos Seimas, 1994-07-07, I-533. Available online: https://e-seimas.lrs.lt/portal/legalAct/lt/TAD/TAIS.5884/lqnVSuumJa (Lietuvos Respublikos 1994).

Subsidiarity in management: Functioning of the principle of subsidiarity is also apparent in a number of management theories. For instance, the theoretician of human relations management—McGregor's (Dartey-Baah 2009)—who interprets management as the art of regulating human relations and, next to theory X (authoritarian), suggests theory Y (democratic attitude towards employees) in which he highlights that humans are inherently creative and active. This reflects the essence of subsidiarity as it facilitates human self-realization—which is one of the highest levels of human social needs—and provides conditions (which we can consider to be a realization of subsidiarity) for the expression of creativity of employees (members of an organization) and the application of knowledge, which in turn results in a significantly greater effect than encouraging employees' physical effort (externally). The only question to answer is what a good manager should do to encourage the expression of a member of the organization, which as such is an expression of the principle of subsidiarity within the organization. This must be the reason this principle is mentioned in the Lithuanian Law on Public Administration as number seven among the other thirteen.

According to another scholar, Drucker (2009, p. 10), each company is made up of people with a variety of abilities, skills and knowledge, who perform many various jobs. It must rely on communication and individual accountability. All its employees need to think about what they want to do and make sure their co-workers know and understand that goal. Everyone has to think about what they have to do for others and make sure that others understand this. Why cannot we view this consideration of employees on what they need to do to make their co-workers understand their goals, as the principle of subsidiarity, though not named by Drucker as such? In our opinion, this is exactly the case. In addition, Drucker emphasized subsidiarity not only in the organization's management activities, but also in the functioning of different public governance bodies, that is, he acknowledged that non-profit organizations are far more effective and admitted that no United States (US) program in the last 40 years that tried to address the social issue by government action provided significant results. However, independent non-profit organizations achieved impressive results, for example, church schools, alcoholics anonymous groups, Samaritans, and so forth and, therefore, the development of autonomous community organizations in the social sector is an important step towards a radical change in government to make it effective again (Drucker 2009, p. 309).

The principle of subsidiarity is even more apparent in G. Peters' theories, because when discussing the importance of organizational culture and internal management he refers to Weber's three sources of authority—traditional, charismatic and rational (traditional and rational authorities are to a large extent based on hierarchy) and draws attention to the fact that hierarchical management has become less practical in the majority of large organizations in the West because, among other cultural values, it has entrenched the right of a person to determine his or her future, whereas workers, especially younger ones, want to participate in decision-making more actively, and this is the case not only in manufacturing companies, but also in public organizations, because the lower levels of these organizations also require new forms of management to be introduced to allow decisions to be made with the participation of those directly affected by the decisions (Peters 2002, p. 96).

Although De Vries uses another term—decentralization—he also supports decision-making at a local level and states that decentralization is likely to increase the opportunities for citizen-oriented policies in order to remove excessive bureaucracy, increase the awareness and sensitivity of officials to local problems. This could result in more efficient implementation of national policies in remote local communities, greater representation of different religious, ethnic and national groups in the policy process, and increased administrative capacity at a local level. This would help to create a structure that could coordinate local projects, resulting in flexible, innovative and creative administration that is more efficiently implemented due to simpler monitoring and evaluation, which can increase political stability and national unity and reduce the scale of inefficiency (De Vries 2012, p. 548). So the questions which address such issues as actions taken to involve employees in decision-making, the attitude towards the decentralization of certain services, increasing administrative capacity in places because it

is more efficient, are directly related to the problem of this research—the functioning of the principle of subsidiarity in practice, that is, in the policy on local self-government.

Whatever perspective we take, the plain fact is that the subsidiarity principle is a quite complex issue that can be scrutinized by many lenses (Drew and Grant 2017). As we have shown above, there are many scientific areas that research subsidiarity such as management, administration studies, political science, philosophy, finance and, last but not least, law.

## 4. Methodology of Research on Subsidiarity in Discourse of Local Self-Government and Local Communities of Lithuania and Poland

The empirical research has been conducted by performing structural content analysis. According to this method, analysis of the content of the selected texts is performed (in this case, documents of the Committee on State Administration and Local Authorities during two sessions (8 and 9) of the Seimas term 2012–2016 and two sessions (1 and 2) of the Seimas term 2016–2020) and the Polish Self-Government and Regional Committee of the Sejm during two years in VII term (2013–2014) and two years in VIII term (2017–2018).

The analysis is performed by using standard measuring instruments (in this case, instances of the use of the word 'subsidiarity') and thus acquiring objective characteristics through procedures (committee deliberations), namely, regarding the fact whether the principle of subsidiarity enshrined in the Law on Local Self-Government (which was introduced into the law by this particular committee) is used when considering law drafts that have an impact on the implementation of the subsidiarity principle in real life.

This method not only identifies the events, facts, and relationships that are actually discussed in the text (the frequency of use of the word 'subsidiarity'), but also traditions and interests (in this case, whether the Seimas Committee, which should serve as a model for other committees and the Seimas, strives to implement the principle of subsidiarity starting with the law-making process and ending with implementation of the law). The essence of the content analysis method is to distinguish certain meaningful units in the text of the document (in this case, the frequency of use of keywords such as subsidiarity, subsidiarity principle), then calculate the frequency of their use and analyse the relationships of different text elements with one another and with the full scope of the information. Recurrence of the subject (subsidiarity, subsidiarity principle) in one or another document proves its significance. In this paper, discourse is understood as an extended thought on the concept of subsidiarity.

The aim of the research is to establish whether the principle of subsidiarity is used in the process of considering documents during the law-making process of the Seimas of the Republic of Lithuania and the Sejm of the Republic of Poland.

The logic behind raising such questions for scientific analysis is that argumentation is used during the law-making process of the Seimas and the Sejm based on certain principles or defined criteria. For instance, the Legal Department of the Chancery of the Seimas reviews compliance of draft laws with the Constitution, laws, principles and rules of drafting laws, whereas the European Law Department under the Ministry of Justice reviews compliance of draft legislation with European Union laws, meanwhile the Seimas committees analyse possible negative consequences of a draft law on criminogenic situation, corruption, or business conditions.

The analysed scientific problem has become even more relevant due to the fact that on 10 June 2016 an amendment of the Article 4 of the Law on Local Self-Government No. I-533 was registered in the Seimas of the Republic of Lithuania. This article was supplemented with Clause 14:

> "14) subsidiarity. Decisions of municipal public administration entities of must be taken and implemented at the most effective level."

The regular committee, which was the Committee on Public Administration and Local Authorities, submitted its conclusion based on the following arguments—the Legal Department of the Chancery

of the Seimas stated that this principle was entrenched by Clause 7 of Article 3 of the Law on Public Administration and was applied to local authorities; thus, it would be excess in this law. The Committee did not approve of such argumentation and pointed out that this principle should be entrenched in the law, taking into account the recommendation of the experts of the Council of Europe, who monitored the implementation of the European Charter of Local Self-Government in Lithuania, submitted during the 22nd Session of the Congress of Local and Regional Authorities of the Council of Europe. The report based on this monitoring recommends the Lithuanian national authorities "to amend Article 4 of the existing Law on Local Self-Government so that the principle of subsidiarity is specifically recognised in the field of local government, by being mentioned as one of its guiding principles".

The Union of Local Community Organizations of Lithuania, the Union of Rural Communities of Lithuania, the Lithuanian Local Action Groups, Lithuanian Social Workers' Association, the Lithuanian Association of Municipal Elders, and the Institute of Business and Rural Development Management of Aleksandras Stulginskis University stated that the law must clearly define the concepts of "participation" and "subsidiarity principle". The Committee rejected the proposal to include the concept of "participation", but approved of inclusion of the concept of "subsidiarity".

On the same day, the committee had deliberations on another amendment to the Law on Local Self-Government, that is, Article 19, which extended the composition and powers of elders. The Union of Local Community Organizations of Lithuania and other listed organizations mentioned the principle of subsidiarity in their argumentation, namely, the necessity to carry out internal decentralization of the local self-government and adopt and implement decisions of municipal public administration entities in accordance with the principle of subsidiarity at the level at which they are most effective, that is, the wards.

Thus, since July 2016, one more principle has been introduced, that is, the subsidiarity principle, along with other principles such as representative democracy, freedom of independence and activity of municipalities, supremacy of the municipal council over accountable executive institutions of a municipality, accountability of executive institutions of a municipality to the municipal council, responsibility before the municipal community, lawfulness of the activities of a municipality and decisions taken by municipal institutions, the adjustment of municipal and state interests when managing public affairs of municipalities, the adjustment of interests of the community and individual residents of a municipality, participation of the residents of a municipality in the management of public affairs of the municipality, transparency of activities, development and activity planning, responsiveness to the opinion of the residents of a municipality, ensuring respect for human rights and freedoms.

In terms of Polish law, the subsidiarity principle has been mentioned only in one place—the preamble of the constitution. It is stated that: Hereby establish this Constitution of the Republic of Poland as the basic law for the State, based on respect for freedom and justice, cooperation between the public powers, social dialogue, as well as on the principle of subsidiarity in the strengthening the powers of citizens and their communities. The main goal of the subsidiarity principle is to give a direction of state organization as well as strengthening the position of citizens in relation to the public administration. However, it is not certain how important this part of the constitution is, due to the fact that there is an open question of whether it is legally binding text or only some kind of guidance. In practical terms it means that the subsidiarity principle should be taken into consideration only with other regulations. This is why it is very often a huge problem to show step by step how this principle should be used in public administration practice. As an example, we can show the constitutional principle of the unitary state. Taking this into consideration, we can state that the Polish regions are nothing more than sub-state governments without any autonomy or independence. It is worth mentioning the fact that the subsidiarity principle was introduced in the Polish constitution as a result of communistic experience, which means that public administration was based on centralization rule. Neither regions nor local units were allowed to deliver any kind of public services on their own. Therefore, it is a slightly different context of introducing the subsidiarity principle than in Western

European countries. The goal of all administration reforms that have been pursuing since 1989 were to empower local communities and strengthening public engagement in decision making process.

We can find more details about the subsidiarity principle in articles 15 and 16 of the Polish constitution. In the first article it is stated that self-government in Poland is based on the decentralization principle. In order to fulfil this principle, self-governments should be equipped with sufficient resources that allow them to deliver public services. In the second article mentioned above, the Polish constitution gives self-government some kind of self-reliance, which is protected by law and courts.

The last part of the Polish constitution that gives more details about self-government is Chapter 7, titled Self-government. There is no doubt that most of the articles stated that there are inspired by the European Charter of Local Government. Therefore, it is no surprise that we can find many articles strictly linked to the Council of Europe standards. In the context of the subsidiarity principle, the most important is the general competence rule that allows local government to deliver all those services that are not assigned to other public administration institutions.

As the main argument in deciding to introduce this principle in the Law on Self-Government was that the subsidiarity principle should be explicitly recognized at the level of local government by entrenching it as one of the basic principles of self-governance, the aim of the current research is to establish to what extent it has become essential or important for the committee that is directly related to the exercise of the powers of self-government and local communities to first of all ensure that the norms of legal regulation providing for decisions to be made at the level that is most effective, which can be proven by the use of the term in the process of considering legal acts, taking into consideration that the norm has been in force for more than a year.

First, it should be noted that the Legal Department of the Seimas, aiming to draw the attention of the members of the Committee that this principle is enshrined in Clause 7 of Article 3 of the Law on Public Administration and is applicable to local self-government, referred to the following wording in the Law on Public Administration: "This principle means that the decisions of entities of public administration must be adopted and implemented at the most efficient level of public administration system", whereas the norm adopted in the Law on Local Self-Government is narrower: "Decisions of municipal public administration entities must be taken and implemented at the most effective level", which means a narrower scope of application of the principle. Thus, although the new supplement to the Law on Local Self-Government does not provide for the principle of subsidiarity in a broader sense, the latter is enshrined in the Law of Public Administration.

For the purpose of the analysis, the Committee on State Administration and Local Authorities was selected as a target committee and documents, considered by the committee, which relate to the activities of municipalities (mainly the Law on Local Self-Government, the Law on Municipal Assets) or documents which contain lots of words that denote functions related to a person (or local community), were chosen on the basis of targeted sampling. (shown as Table 1)

**Table 1.** The analysis of the documents of the Committee on State Administration and Local Authorities (CSALA) during sessions 8 and 9 of the Seimas term 2012–2016[2].

| No, Session, Date of Deliberation | Legal Act (the Analysed Explanatory Notes and Minutes of the Conclusions of the Deliberations of the Committee). | Content of the Proposal | Submitted by. Are There Any Arguments that the Opinion under Consideration Is Based on the Principle of Subsidiarity? |
|---|---|---|---|
| 1. 10 June 2016, Regular session No 8 | Amendment of the Articles 13, 15 and 27 of the Law on Local Self-Government | The question on loss of the mayor's mandate before the term of office expires, removal of the mayor from office and impeachment. | CSALA regular. No argumentation provided. |
| 2. 15 June 2016 | Amendment of the Articles 19 and 24 of the Law on Local Self-Government | The mayor is eligible to have public advisers during the term of office. | CSALA regular. No argumentation provided. |

**Table 1.** *Cont.*

| No, Session, Date of Deliberation | Legal Act (the Analysed Explanatory Notes and Minutes of the Conclusions of the Deliberations of the Committee). | Content of the Proposal | Submitted by. Are There Any Arguments that the Opinion under Consideration Is Based on the Principle of Subsidiarity? |
|---|---|---|---|
| 3. 16 June 2016 | Amendment of the Articles 4, 10, 11, 14, 26 of the Law on Associations of Owners of Multi-Apartment Residential Buildings and other Types of Buildings | The members of the Community may, in accordance with the procedure laid down in the Statutes of the Community, express their views on the matters discussed at the meeting in writing in advance. | CSALA additional. No argumentation provided. |
| 4. 22 June 2016 | Amendment of the Articles 4 and 9 of the Law on Local Self-Government and Supplement to Article 9(1) of the Law | To lay down the requirements that the local authorities must observe when provision of public services is related to performance of economic activities. | SCALA additional. No argumentation provided. |
| 5. 23 June 2016 | Amendment of the Articles 3, 4, 6, 9, 10(3), 13, 14, 15, 16, 20, 23, 29, 31, 32, 33, 34, 35, 50 of the Law on Local Self-Government | The requirement of at least 5 per cent of the residents to participate in the elections of the elders, the procedure for admission of wards providing for an exception when the ward is elected to be a state politician. | SCALA regular. No argumentation provided. |
| 6. 23 June 2016 | Amendment of the Articles 13, 15 and 27 of the Law on Local Self-Government | The provision on decision of the municipal council on impeachment does not comply with the Law on Local Self-Government as well as with the Law on Elections to Municipal Councils. | SCALA regular. No argumentation provided. |
| 7. 15 September 2016 Regular session No 9 | Amendment of the Articles 11, 16, 20 and 29 of the Law on Local Self-Government | Proposal to clarify the provisions on appointment of Deputy Mayor(s) and dismissal thereof before the term of office expires | SCALA regular. No argumentation provided. |
| 8. 15 September 2016 | Amendment of the Articles 11, 13, 20 and 29 of the Law on Local Self-Government | The function of the mayor of the municipality is to present, coordinate and submit nominations of the chairperson of the Control Committee. | SCALA regular. No argumentation provided. |
| 9. 15 September 2016 | Amendment of the Article 2 of the Law on Temporary Direct Rule On The Municipal Territory | Introduction of temporary direct management in the territory of the municipality. | SCALA regular. No argumentation provided. |
| 10. 21 September 2016 | Amendment of the Articles 3, 12, 14, 16, 20, 27, 28 and 30 of the Law on Local Self-Government | Replacement of the term "Municipal Controller (Municipal Control and Audit Service)" by the term "Municipal Control and Audit Service" and introduction of two positions of auditors. | SCALA regular. No argumentation provided. |

**Table 1.** *Cont.*

| No, Session, Date of Deliberation | Legal Act (the Analysed Explanatory Notes and Minutes of the Conclusions of the Deliberations of the Committee). | Content of the Proposal | Submitted by. Are There Any Arguments that the Opinion under Consideration Is Based on the Principle of Subsidiarity? |
|---|---|---|---|
| 11. 21 September 2016 | Amendment of the Articles 14 and 15 of the Law on Local Self-Government and Supplement to Article 19 of the Law | A proposal to impose an obligation on the minority (opposition) of the municipal council to carry out democratic political supervision of the activities of the majority of the municipal council. | SCALA regular. No argumentation provided. |
| 12. 3 November 2016 | Amendment of the Article 3 of the Law on Methodology of Determination of Municipal Budget Revenue | Funds to be allocated to the municipal budget are counted by means of the mechanism of the levelling of differences in the income tax of individuals of municipalities (Article 7) and the mechanism of the levelling of differences in the expenditure structure (Article 8). | SCALA additional. No argumentation provided. |

Source: Own work.

The analysis included the documents of the Committee on State Administration and Local Authorities during two sessions (8 and 9) of the Seimas term 2012–2016 and two sessions (1 and 2) of the Seimas term 2016–2020.

The analysis of these documents revealed that, during deliberations, the following principles were used in the argumentation—responsible governance, equality of the council members, the constitutional principle of equality, clarity, systematicity, equal treatment, freedom of movement, representative democracy, autonomy, freedom of action, proportionality, and publicity. (shown as Table 2)

**Table 2.** The analysis of the documents of the Committee on State Administration and Local Authorities (CSALA) during sessions 1 and 2 of the Seimas term 2016–2020.

| No, Session, Date of Deliberation | Legal Act (the Analysed Explanatory Notes and Minutes of the Conclusions of the Deliberations of the Committee). | Content of the Proposalo | Submitted by. Are There Any Arguments that the Opinion under Consideration Is Based on the Principle of Subsidiarity? |
|---|---|---|---|
| Regular session No 1 1. 2 December 2016 | Amendment of the Articles 4 and 9 of the Law on Local Self-Government and Supplement to Article 9(1) of the Law | On delay of entrance into force of Article 9(1) of the Law No XII-2741 for half a year to ensure the necessary legal preconditions to be provided for preparation and implementation of new provisions of the law which regulate the economic activity of municipalities. | SCALA regular. No argumentation provided. |
| 2. 15 December 2016 | Draft Law on Remuneration of Employees of State and Municipal Institutions | Determining the terms of remuneration of employees not covered by Article 1 (2) under this law. | SCALA additional. No argumentation provided. |

---

2    Documents of the Committee on State Administration and Local Authorities in Lithuania (CSALA) during sessions 8 and 9 of the Seimas term 2012–2016. Available online: https://www.lrs.lt/sip/portal.show?p_r=8934&p_k=2 (accessed on 14 November 2019).

**Table 2.** *Cont.*

| No, Session, Date of Deliberation | Legal Act (the Analysed Explanatory Notes and Minutes of the Conclusions of the Deliberations of the Committee). | Content of the Proposalo | Submitted by. Are There Any Arguments that the Opinion under Consideration Is Based on the Principle of Subsidiarity? |
|---|---|---|---|
| 3. 15 December 2016 | Amendment of the Article 39 of the Law on Elections to Municipal Councils | On the increase of the election deposit in elections to municipal councils from two to ten AMW. | SCALA additional. No argumentation provided. |
| 4. 15 December 2016 | The Republic of Lithuania Draft Law on Family Card. | The possibility for large families to buy certain services and goods cheaper. | SCALA additional. No argumentation provided. |
| 5. 21 December 2016 | Amendment of the Articles 4 and 9 of the Law on Local Self-Government and Supplement to Article 9(1) of the Law | Prevention of potentially non-transparent and economically unjustified municipal economic activities by establishing that municipalities set up a new legal entity or entrust the implementation of new economic activities to the existing legal entities owned by the municipality | SCALA regular. No argumentation provided. |
| 6. 28 February 2017 | Amendment of the Article 23 of the Law on the Management, Use and Disposal of State and Municipal Assets | Seeking to transfer the functions of the management coordination centre performed by the manager of centrally managed state assets to the public institution "Monitoring and Forecasting Agency". | SCALA additional. No argumentation provided. |
| 7. 31 March 2017 Regular session No 2 | Amendment of the Article 11 of the Law on State and Municipal Enterprises | Determining that the same person may be appointed as the manager of the same enterprise for not more than two consecutive terms. | SCALA additional. No argumentation provided. |
| 8. 19 April 2017 | Amendment of the Article 22 of the Law on Land Reform | By 1 January 2018, the Director of the Municipal Administration has to issue an order to approve the list of the urban areas where new land plots would be formed with the aim to transfer them into ownership to citizens without payment to ensure restoration of their ownership rights to urban land. | SCALA additional. No argumentation provided. |
| 9. 27 April 2017 | Amendment of the Articles 14 and 15 of the Law on Local Self-Government and Supplement to Article 19 of the Law | The minority (opposition) of the municipal council has a right to delegate its members to serve as the Chair of the Control Committee, the Chair of the Ethics Committee, and the Chair of the Anti-Corruption Committee. Chairs and deputy chairs of the committees, except the Control Committee, are elected by the committees on the proposal of the mayor. | SCALA regular. No argumentation provided. |

**Table 2.** *Cont.*

| No, Session, Date of Deliberation | Legal Act (the Analysed Explanatory Notes and Minutes of the Conclusions of the Deliberations of the Committee). | Content of the Proposalo | Submitted by. Are There Any Arguments that the Opinion under Consideration Is Based on the Principle of Subsidiarity? |
|---|---|---|---|
| 10. 3 May 2017 | Amendment of the Article 14 of the Law on Local Self-Government | The Control Committee should report to the Municipal Council on its activities at the beginning of each year. | SCALA regular. No argumentation provided. |
| 11. 5 May 2017 | Amendment of the Article 26 of the Law on Local Self-Government | The possibility for municipal councilors to use the allowance to pay for rental of office premises. | SCALA regular. No argumentation provided. |
| 12. 31 May 2017 | Amendment of the Articles 3, 14, 15, 16, 31, 34 and 35 of the Law on Local Self-Government | Providing local communities, their representatives and community organizations with more rights in dealing with local (municipal) affairs. | SCALA regular. No argumentation provided. |
| 13. 31 May 2017, 22 May 2017, 15 May 2017 | Amendment of the Articles 4 and 9 of the Law on Local Self-Government and Supplement to Article 9(1) of the Law | Municipalities have no obligation to apply to the Competition Council for permission to outsource public services to a legal entity under their control. | SCALA regular. No argumentation provided. |
| 14. 1 June 2017 | Amendment of the Articles 16 and 27 of the Law on Local Self-Government | Municipal controllers provide conclusions and the municipal council makes decisions on the appropriateness of the partnership project. | SCALA regular. No argumentation provided. |
| 15. 1 June 2017, 22 June 2017, 28 June 2017 | Amendment of the Article 15 of the Law on Regional Development | The criteria for formation of the Regional Development Council include participation of representatives of entities not subordinate to the Government or ministries in the existing regional development councils. | SCALA regular. No argumentation provided. |
| 16. 8 June 2017 | Amendment of the Article 39 of the Law on Elections to Municipal Councils | The election deposit is to be returned 60 days after the official announcement of the final results of the election. | SCALA regular. No argumentation provided. |
| 17. 22 June 2017 | Amendment of the Articles 12 and 15 of the Law on State and Municipal Enterprises | Establishing an exception for the calculation of the return on equity of state-owned enterprises which carry out maintenance of roads of state significance. | SCALA additional. No argumentation provided. |
| 18. 3 July 2017 | Amendment of the Articles 6 and 7 of the Law on Local Self-Government | Management of municipal spatial data set. | SCALA regular. No argumentation provided. |
| 19. 3 July 2017 | Amendment of the Article 6 of the Law on Local Self-Government | Maintenance, repair, and construction of roads and streets and organization of traffic safety in gardeners' communities. | SCALA regular. No argumentation provided. |

Source: Own work.

The analysis of these documents revealed that during deliberations the following principles were used in the argumentation—transparency, equality, appreciation, proportionality, public interest, competitiveness, reconciling the interests of local authorities and the state. (shown as Table 3)

**Table 3.** The analysis of the documents of the Self-Government and Regional Policy Committee during the VII term of the Sejm in the years 2013–2014.

| No, Session, Date of Deliberation | Legal Act (the Analyzed Explanatory Notes and Minutes of the Conclusions of the Deliberations of the Committee). | Content of the Proposal | Submitted by. Are There Any Arguments that the Opinion under Consideration Is Based on the Principle of Subsidiarity? |
|---|---|---|---|
| 1. No. 123 in 6 March 2013. | Public display and discussion about Self-government Assessment Report prepared by prof. prof. Huberta Izdebskiego and Jerzego Hausnera | No proposal | During Prof. H. Izdebski (expert) presentation, he shown basic dysfunctionalities of the Polish self-government. One of them was growing centralization process of public service delivery that is in contradiction with subsidiarity principle. In his view, this centralization process has resulted in growing importance of the state in public administration. |
| 2. No. 178 in 11 September 2013 | Discussion about "State Development Strategy 2020" prepared by Prime Minister (paper No 972) as well as Public Information about implementation "State Development Strategy 2007-2015 in 2011" (paper No 737). | No proposal. That was only the explanation of doubts raised by one of Member of Parliament during the discussion. | Marceli Niezgoda (Udersecretary of State in the Ministry of the Regional Development) explained that the Strategy took into account further decentralization in regional policy based on subsidiarity principle. |
| 3. No. 235 in 21 January 2014 | Discussion over the Subcommittee report devoted to governmental blueprint act—Amendment to the Principles of Regional Development Act and Other Acts (paper No 1881). | One of the participants blame the government that presented blueprint stood in contradiction with subsidiarity principle. | Krzysztof Szczerski, MP (Law and Justice) claimed that substitution of regional contract to the territorial contract undermined subjectivity of regional government. |

Source: Own work.

The analysis of these documents revealed that during deliberations the following principles were used in the argumentation—as a constitutional principle guaranteeing that local governments delivery a vast majority of public services; as an important rule that completes decentralization (shown as Table 4), therefore it creates solid foundation to leave more public money in local administration and last but not least as a basic principle in political system in Poland that should be understand as a mutual relation between central and local government based on partnership.

**Table 4.** The analysis of the documents of the Self-Government and Regional Policy Committee during the VIII term of the Sejm in the years 2017–2018[3].

| No, Session, Date of Deliberation | Legal Act (the Analyzed Explanatory Notes and Minutes of the Conclusions of the Deliberations of the Committee). | Content of the Proposal | Submitted by. Are There Any Arguments that the Opinion under Consideration Is Based on the Principle of Subsidiarity? |
|---|---|---|---|
| 8th of March 2017 | Governmental information about the implementation of Public Legal Assistance and Legal Education Act 2015 | One of the MP accused the government to overruled public legal assistance which was in contradiction with subsidiarity principle known as fundamental principle in EU. In the light of a lack of flexibility, the role of county government is to execute legal regulations rather than do it in decentralize way. | Partly |
| 4th of July 2017 | Governmental Amendment Act to the Principles of Cohesion Policy Programme Act in 2014-2020 Perspective and Other Acts (Paper No 1636). | During the discussion about the amendment introducing more legal power for voivode (government administration in regions) over the structural fund, one of the MP stood up against this governmental proposal arguing that it is against subsidiarity principles. From his point of view it would decrease the power of self-government to manage structural funds. | No |

**Table 4.** *Cont.*

| No, Session, Date of Deliberation | Legal Act (the Analyzed Explanatory Notes and Minutes of the Conclusions of the Deliberations of the Committee). | Content of the Proposal | Submitted by. Are There Any Arguments that the Opinion under Consideration Is Based on the Principle of Subsidiarity? |
|---|---|---|---|
| 26th of October 2017 | Governmental Amendment Act to the Collective Supply of Water and Draining the Sewerage Act (Paper No 1905). | During the debate about the proposal allowing government to take control of charging process, one of the self-government national association representative emphasized that it is against subsidiarity principle. His statement was supported by all organisations representing self-government in Central and Local Common Commission. In addition to that, he figured but that proceeding amendment broken constitution, especially in terms of decentralization rule. | No |
| 28th of February 2018 | Governmental information about the consequence of division the Mazoviecki Region into two separate statistical units | Regarding discussion over governmental information about new statistical units, participants exchanged views about regional policy in the UE. During this conversation Marshal of Mazoviecki Region describe subsidiarity principle as one that gives people feeling about the UE on daily basis. http://orka.sejm.gov.pl/zapisy8.nsf/0/ D8A56AD99458E653C1258247004F87AB/ %24File/0289808.pdf | Yes |
| 7th of June 2018 | Subcommittee Report about proposal of the President of Poland in terms of Public Legal Assistance and Legal Education Amendment Act (Paper No 1868). | During the discussion over proposed Amendment Act, Janusz Sanocki (independent MP) said it was too complicated and braking basic principles such as subsidiarity. In his point of view, new legal instruments left a lack of flexibility for local government and as a result introduced strict rules how to implement it. He was in favor of much more general framework that gave local government room to choose the best way of implementation. http://orka.sejm.gov.pl/zapisy8.nsf/0/ F384B2B60EB453AEC12582B10049320C/ %24File/0322508.pdf | No |
| 27th of June 2017 | Governmental Amendment Act of Specific Solutions in therms of Ostowice Municipality in Westpomerania Region (Paper No 2650). | Deputy Minister of the Internal Affairs and Public Administration P. Szafernaker said it was very unpleasant situation for both sides to liquidate one of the municipality (Ostrowice). However, both sides agreed that it was the best option due to the financial situation. In his mind, this was a good example of subsidiarity principle defined as state intervention as local government is unable to solve their own problems. | Yes |

Source: Own work based on Kancelaria Sejmu Biuro Komisji Sejmowych (2017a, 2017b, 2017c, 2018).

The analysis of these documents revealed that during deliberations the following principles were used in the argumentation—bigger flexibility for local government to deliver public services, subsidiarity principle as a solid foundation of the European policies, and last but not least local independence as long as it is able to work on their own.

---

[3]    Documents of the Self-Government and Regional Policy Committee in Poland during the VIII term of the Sejm in the years 2017–2018.

## 5. Conclusions

According to the conducted research we are able to formulate the following conclusions from both the Lithuanian and Polish perspectives. Firstly, the concept of subsidiarity, which has withstood the challenges of time in the Christian, economic, and political discourse, has been analyzed in the Lithuanian discourse for some time now; thus, it can be said that its meaning can be fully clear to any user of the word.

Secondly, since the object of this research, namely, local self-government and local community, falls within the domain of public administration based on management theoreticians, the application of the principle of subsidiarity is fully appropriate to their activity, which is also confirmed by the fact that this principle is enshrined in the Law of Public Administration of the Republic of Lithuania and the Law of Local Self-Government of the Republic of Lithuania, along with other principles.

Thirdly, the content analysis of the legal acts considered by the Committee on State Administration and Local Authorities revealed that the term "subsidiarity," even after being entrenched along with other principles of local self-governance in the Law on Local Self-Government in 2016, is not used; thus, it has merely a declarative effect in the process of making decisions related to local self-governance and local community, that is, the decentralization of management.

Fourthly, the Polish experience in using the subsidiarity principle seems to have quite a different dimension for several reasons. The first one is that the subsidiarity principle is mentioned only in one place in Polish law, it is a preamble to the constitution. The second one is that the subsidiarity principle is very often mentioned during general debates about the position of self-governance in public administration in general. Moreover, in most cases, the subsidiarity principle is mentioned by local government representatives in order to show how central government is trying to take control of public services or decrease the level of self-reliance.

Last but not least, taking the Polish cases into consideration, the clear truth is that the subsidiarity principle has more normative meaning rather than operational application. Therefore, it is more likely that it will be used as a rhetorical argument during the public and/or legislative debate rather than as a useful tool employed in the public management process. From our perspective, it is highly recommended to think more practically about this principle and put much more effort into looking for a deeper meaning of subsidiarity.

Taking both Lithuanian and Polish experiences into consideration, we can highlight the following conclusions. In Lithuania, the subsidiarity principle is much more frequently a subject of discussion than in Poland. Furthermore, Lithuanian politicians are talking about this subject in the context of legal amendments, whilst Polish members of parliament only use the subsidiarity principle in the context of political dispute as one argument. However, there is one similarity that we can observe in terms of this principle. In most cases, both in Lithuania and in Poland, subsidiarity is associated with political issues. There is a lack of understanding of this principle in the context of public administration management or the financial independence of local authorities. The reason for doing so might be a good assumption of the next research.

**Author Contributions:** Writing—original draft, R.G., S.N. and K.M. All authors have read and agreed to the published version of the manuscript.

**Conflicts of Interest:** No potential conflict of interest was reported by the authors. This work was not supported by any external source. The authors take all responsibility to prepare and conduct research.

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
