# Peer review of "Subsidiarity Principle—Its Realization in Self-Government in Lithuania and Poland"

_admsci, doi:10.3390/admsci10010014_

Round 1

Reviewer 1 Report

Hello,

Thank you for researching this topic and I am happy to peer review this article. This article is interesting and there are my comments.

Minor Issues

There are very few new references in the article. As we are in year 2020 it is more relevant to refer and cite article between 2014-2020, unless the reference articles are classic articles to subsidiarity; self-government; public administration etc. In the reference there are no new articles referred after 2017. There are gov: document reference from 2018 but no research article.

Future research/study needs to be added and it is missing from the article. English language and style are fine/minor spell check required. Format of the article especially reference to be formatted in a research style

Author Response

Hello, 

Thank you very much for your comments. Actually, I agree with them entirely. Therefore, I have added some latest literature about the subsidiarity principle to the paper. 

Best wishes,

RG

Reviewer 2 Report

The paper offers interesting insights both on the concept of subsidiarity and on its application in Lithuania and Poland. At the general level, I think the text might be publishable on Administrative Sciences but it would need a considerable revision process.

I think there are three areas where the paper can be improved:

The article requires English editing.For example, sometime the author/s do not correctly use the article (or omit it) before substantive names introduced for the first time. Moreover, several sentences in the manuscript are not clearly written.

Conceptual clarity. Sometime the theoretical analysis especially at the beginning of the paper is not very clear. I provide two examples:

- At page two I found the point on Hegel and the theoretical discussion unclear. Could you please specify more what you mean? I understood the concept is not new but this could be spelled out more clearly.

- Still at page two the author/s claim:

“The authors of the article have observed this tendency due to their long experience in developing qualifications of civil servants and politicians of Lithuania and Poland”.

The authors should substantiate the point with some evidence (from the literature or empirical) rather than just say they observed it in their long working experience.

The paper would benefit from the insights of international literature on the concept of subsidiarity but also on cognate literatures that although not mentioned in the paper have been often discussed in relation to subsidiarity. I refer to social policy (and in particular welfare regime literature) and social capital literature.

On the concept of subsidiarity in Europe:

- Van Kersbergen, K. and Verbeek B. (1994) The Politics of Subsidiarity in the European Union, Journal of Common Market Studies 32 (2): 215-236.

- Linking social capital and subsidiarity in European regional policy:

Leonardi, R. and Paraskevopoulos, C. (1996) Social Capital and Learning Institutional Networks: Making Sense of Subsidiarity in European Regional Policy

- The cases of Poland and Lithuania should be placed within the public policy/social policy literature. I suggest to refer to this article:

Aidukaite, J. (2009) Old Welfare State Theories and New Welfare Regimes in Eastern Europe: Challenges and Implications. Communist and Post-Communist Studies, 42 (1): 23-39.

- In order to embed the concept of subsidiarity (which is particularly important within conservative welfare regimes) within welfare regime theory, I suggest the author/s also engage with the following works:

Esping-Andersen, G. (1990) The Three Worlds of Welfare Capitalism. Princeton: Princeton University Press.

Ferragina, E. and Seeleib-Kaiser, M. (2011) Welfare regime debate: past, present and futures? Policy & Politics 39 (4): 583-611.

Ferragina, E. (2017) The Welfare State and Social Capital in Europe: Reassessing a Complex Relationship. International Journal of Comparative Sociology, 58 (1): 55-90.

In order to reflect on how social capital, subsidiarity and communities are related:

Putnam, R.D. (1993) Making Democracy Work. Civic Traditions in Modern Italy. Princeton: Princeton University Press.

Putnam, R.D. (2000) Bowling Alone. The Collapse and Revival of American Society. New York: Simon and Schuster.

Ferragina, E. and Arrigoni, A. (2017) The Rise and Fall of Social Capital: requiem for a theory? Political Studies Review, 15 (3): 355-367.

On the use of a regional perspective to better methodologically place your cases:

Ferragina, E. (2009) Social Capital and Equality: Tocqueville’s legacy. LIS Working Paper Series 515

Ferragina, E. (2012) Social Capital in Europe: A Comparative Regional Analysis. Cheltenham: Edward Elgar.

Author Response

Dear Sir or Madam, 

Thank you for your revision of our paper. We find all comments very useful and in most cases, we implement proposed changed. 

  1. The text has been carefully read and language changes have been implemented.
  2. The sentence: “The authors of the article have observed this tendency due to their long experience in developing qualifications of civil servants and politicians of Lithuania and Poland” - has been removed due to the fact that it is hard to give straightforward references both in Lithuania and in Poland.
  3. The results of the research have been extended by showing comparative conclusions and showing new areas of research. 
  4. New literature positions have been added. 

One again, thank you very much for your review. Afterword, we find our paper much better than before. 

Yours sincere,

RG